# Ammunition Waste Pollution and Preliminary Assessment of Risks to Child Health from Toxic Metals at the Greek Refugee Camp Mavrovouni

**DOI:** 10.3390/ijerph191610086

**Published:** 2022-08-15

**Authors:** Katrin Glatz Brubakk, Elin Lovise Folven Gjengedal, Øyvind Enger, Kam Sripada

**Affiliations:** 1Department of Psychology, Out-Patient Clinic for Children and Adolescents, Norwegian University of Science and Technology (NTNU), NO-7034 Trondheim, Norway; 2Faculty of Environmental Sciences and Natural Resource Management, Norwegian University of Life Sciences (NMBU), NO-1432 Aas, Norway; 3Centre for Global Health Inequalities Research (CHAIN), Norwegian University of Science and Technology (NTNU), NO-7491 Trondheim, Norway

**Keywords:** migrant health, refugee health, children’s environmental health, health impact assessment, heavy metals, environmental toxicants, developmental toxicity, Lesbos, Kara Tepe, Greek islands, Λέσβος, ΚΥΤ Μόριας, Μαυροβούνι

## Abstract

The Mavrovouni refugee camp near the former Moria camp on the island of Lesvos, Greece, housed approximately 3000 asylum-seekers including children as of October 2021. The camp was built on the site of a military shooting range. This study aimed to characterize the soil contaminants and assess the risk of toxic environmental exposures for children living in Mavrovouni. Methods: Samples of surface soil (0–2 cm depth; particle size < 2 mm) from eight locations inside the camp were compared with two reference samples. Soil samples were microwave digested using a mixture of nitric and hydrofluoric acids and analyzed for lead (Pb), antimony (Sb), bismuth (Bi), and other metals using inductively coupled plasma mass spectrometry. These values were compared with action limits established by the Norwegian Environment Agency for kindergartens, playgrounds, and schools. Findings: Five of eight soil samples from inside the camp exceeded Pb levels of 100 mg/kg, which is currently the maximum acceptable value of Pb in soil for playgrounds in Norway. Two sites had extreme soil Pb levels of approximately 8000 mg/kg and 6000 mg/kg. The concen-tration of Sb and Bi in the surface soil of the firing range area strongly indicated environmental contamination, most likely from previous military activity and ammunition residue that has re-mained on the surface soil. Concentrations of arsenic (As), cadmium (Cd), copper (Cu), and zinc (Zn) in surface soil were lower than action limits. Discussion: Extremely high levels of Pb, together with high levels of Sb and Bi, were identified in soil where children live and play in the Mavrovouni refugee camp. This is the first independent study of environmental contamination at this camp and adds to the limited evidence base documenting Pb exposures prior to migrant and refugee reset-tlement. On top of the multiple existing public health crises and traumas that these asylum-seeking families face, exposure to toxic ammunition residues may have profound impacts on children’s development and health for years to come.

## 1. Introduction

Following a deadly fire in September 2020, 13,000 children and their families in the Greek refugee camp, Moria, were relocated to the nearby Mavrovouni site. Here, temporary shelters for asylum-seekers were built directly on a disused military shooting range located outside Mytilene, the capital of the island, Lesvos. As of October 2021, around 3000 people were living in the Mavrovouni Temporary Reception and Identification Centre (TRIC), of whom approximately one-third were children. UNICEF and other humanitarian actors called for all children to be transferred off Lesvos, in the context of inadequate living conditions and persistent mental health crises in the camps.

According to the Greek government, the Mavrovouni site had operated as a firing range field from 1926 until the day it was taken by the Ministry of Migration and Asylum, after the Moria fire [1]. Firing ranges, which are typically used by adults on an intermittent basis, have dangerous concentrations of lead (Pb) and high levels of other toxic metals such as antimony (Sb) and bismuth (Bi) [2,3,4,5,6]. Soil has been identified as an important pathway for children’s Pb exposure [7,8].

There is no known safe level of exposure to Pb [9] and children are especially sensitive to exposure to toxic chemicals during their early years and in utero [10]. Child-specific behaviors like crawling and hand-to-mouth activity mean that children are proportionately more exposed to the environment than adults [11,12]. Even low levels of exposure to Pb disrupts the neurodevelopment that occurs during the early years, which forms the basis for lifelong health and brain function [13]. Childhood Pb exposure is linked to a significant drop in IQ, permanent psychological problems, hearing loss, and cardiovascular disease, among other health effects [14,15]. Therefore, Pb poisoning remains a major health burden for children [16,17].

Previous studies after resettlement showed that many refugee children have been exposed to Pb [18,19,20]. However, there is limited research documenting Pb exposures prior to resettlement, and specifically inside refugee camps.

The Mavrovouni site has not been comprehensively studied for contamination after being repurposed as a refugee camp, beyond one report based on sampling at the site by the Hellenic Geological and Mineral Research Authority (EAGME) [21]. New migrant children in Greece may be more likely to have elevated blood Pb levels compared with native Greek children [22], which makes it important to understand the environmental risks encountered in refugee camps. Given the high proportion of children living on Lesvos and their precarious living situations [23], an independent study is therefore warranted to inform public health action.

This is the first independent environmental monitoring study of the Mavrovouni refugee camp site on Lesvos.

## 2. Materials and Methods

### 2.1. Sampling of Soil

A simple manual method for the sampling of surface soil was chosen. Material between the ground surface to about 2 cm depth was collected by means of a stainless-steel spoon, transferred to a 100 mL polypropylene container, and sealed with the associated screw cap. Collection of eight samples inside the camp took place on 10 October 2020, and two reference samples were collected just outside the camp on 16 November 2020. Location of sampling points was determined with a purposive approach based on where children in the camp spend their time (Figure 1). Samples were transported to the Norwegian University of Life Sciences (Ås, Norway) for analysis.

### 2.2. Chemical Analysis

Chemical analysis was performed at the Norwegian University of Life Science, Faculty of Environmental Science and Natural Resource Management. In the laboratory, the soil samples were heated to 120 °C for 24 h in a heating cabinet (Termaks, Bergen, Norway), according to routines for infection control and to prevent the spread of non-native species. Approximately 0.250 g accurately weighed (Sartorius type 1801 analytical balance, Sartorius AG, Göttingen, Germany) samples of sieved soil (2 mm) were microwave digested (Milestone UltraClave 3, Milestone Srl, Sorisole, Italy); hold time 40 min at the maximum temperature 260 °C. Due to different chemical compatibility among elements, to one sample series of soil was added 5.00 mL ultra-pure HNO_3_ (69% weight (*w*)/*w*, sub-boiled ultra-pure), whereas to a parallel series was added 5.00 mL HNO_3_ (69% *w*/*w*, sub-boiled ultra-pure) and 1.00 mL HF (48% *w*/*w*, sub-boiled ultra-pure). Soil, procedural blanks, and reference materials were digested in Teflon tubes (Milestone Srl, Sorisole, Italy) and subsequently quantitatively transferred into polypropylene centrifuge tubes (Sarstedt, Nümbrecht, Germany), and finally diluted to 50 mL with deionized water. The nitric acid (Merk KgaA, Darmstadt Germany) and hydrofluoric acid (VWR Chemicals BDH^®^, VWR International, Radnor, PA, USA) used were purified by distillation using Milestone duoPUR (Milestone Srl, Sorisole, Italy) and Savillex DST-1000 (Savillex, Eden Prairie, MN, USA), respectively. In sample preparation, a 100–5000 µL electronic pipette (Biohit, Helsinki, Finland) was used. Standard reference materials (SRM) and method blank samples were prepared in the same manner as the respective sample matrices. Deionized water (>18 MW) and reagents of analytical grade or better were used throughout.

The digested soil samples were analyzed for the content of arsenic (As), Bi, cadmium (Cd), copper (Cu), Pb, Sb, uranium (U), and zinc (Zn). Quantification of the total element concentrations was conducted with inductively coupled plasma mass spectrometry using an Agilent 8800 Triple Quadrupole (Triple Quadrupole Inductively Coupled Plasma Mass Spectrometer; Agilent Technologies, Hachioji, Japan) using different reaction modes (oxygen (O_2_) and helium (He)). The masses were (Q1/Q2): As (75/91), Cd (111/111 and 114/114), and Sb (121/137) with gas mode oxygen (O_2_). Regarding Bi, Cu, Pb, Zn, and U, the total element concentrations were determined using helium (He) collision. To ensure metrological traceability and to check for accuracy, three standard reference materials were used: NIST SRM 2710a: Montana I Soil, highly elevated trace element concentrations, NIST SRM 2711a: Montana II Soil, moderately elevated trace element concentrations (National Institute of Standards & Technology, Gaithersburg, MD, USA.), and NCS ZC 73,007 Soil (NCS Testing Technology Co., Ltd., Beijing, China). The mixture of nitric acid and hydrofluoric acid was found to give the best compliance with the reference materials. In general, the obtained data were within 95% confidence level or showed <15% bias compared with the certified values issued (Appendix A). The within-laboratory reproducibility (RSD) for Cu, Sb, Pb, and Bi in the SRMs ranged from 0.0–7.3% (*n* = 4) (Appendix A). The detection and quantification limits were low compared with concentrations measured in the soil samples (Table 1, Appendix A).

### 2.3. Statistical Analysis

Average surface soil concentration and associated standard deviation (mg/kg), as well as the relative standard deviation (RSD, %) was calculated for each sampling site, for each of the five metals of interest. Average concentrations were compared across the sampling sites. The descriptive statistical analysis was performed in Microsoft Excel. Principal component analysis (PCA) conducted in XLSTAT [24] was used to look for trends and commonalities in the data that could be related to the properties of the different soil samples.

**Table 1 ijerph-19-10086-t001:** Average concentration of copper (Cu), antimony (Sb), lead (Pb), bismuth (Bi), and uranium (U) in surface soil (0–2 cm) collected in the Greek refugee camp Mavrovouni. N indicates the number of replicates. The color code specifies the action limits with respect to concentration of Pb in soil in kindergartens, playgrounds, and schools according to the Norwegian Environment Agency [25]. Green: 100 mg/kg—Quality criteria normal. Yellow: 100–300 mg/kg—No action indicated. Orange: 300–700 mg/kg—Immediate action should be considered and the site prioritized in the action phase. Red: 700–2500 mg/kg—Immediate action should be taken. See Figure 1 for location of sampling sites 1–8 and reference sites Ref 1 and Ref 2.

		Pb		Sb		Bi		Cu		U	
Site #	N	Average ± SD (mg/kg)	RSD (%)	Average ± SD (mg/kg)	RSD (%)	Average ± SD (mg/kg)	RSD (%)	Average ± SD (mg/kg)	RSD (%)	Average ± SD (mg/kg)	RSD (%)
**1**	7	46 ± 2.2	4.8	1.2 ± 0.13	11	0.17 ± 0.014	8.3	40 ± 1.5	3.7	1.5 ± 0.16	11
**2**	7	150 ± 44	29	1.5 ± 0.12	7.7	0.19 ± 0.010	5	39 ± 1.4	3.6	2.3 ± 0.14	6.1
**3**	7	(8 ± 3.2) × 10^3^	41	(1.6 ± 0.65) 10^2^	40	2.0 ± 1.1	55	240 ± 63	27	3.0 ± 0.15	4.8
**4**	7	590 ± 32	5.4	9.2 ± 0.83	9.1	0.26 ± 0.036	14	92 ± 8.9	10	2.0 ± 0.13	6.7
**5**	1	6.9		0.49		<0.03		6.7		0.73	
**6**	7	41 ± 1.1	2.6	1.3 ± 0.076	5.7	0.23 ± 0.008	3.2	43 ± 2.3	5.4	2.7 ± 0.076	2.8
**7**	7	(6 ± 10) × 10^3^	192	(2 ± 47) 10^2^	215	3.3 ± 7.8	233	100 ± 22	23	2.4 ± 0.15	6.3
**8**	7	240 ± 48	20	2.5 ± 0.18	7.1	0.25 ± 0.013	5	44 ± 2.0	4.5	3.1 ± 0.13	4.4
**Ref 1**	7	18 ± 2.9	17	1.5 ± 0.18	12	0.058 ± 0.029	51	23 ± 2.5	11	0.8 ±0.12	13
**Ref 2**	1	13		1.4		0.074		21		1.3	

Abbreviations: RSD, relative standard deviation.

### 2.4. Reference Values

Soil element concentrations were compared with the action limits set by the Norwegian Environment Agency with respect to the concentrations of contaminants in soil in kindergartens, playgrounds, and schools [25]. These health-based action limits were derived from a health assessment given by the Norwegian Institute of Public Health [26]. Action limits are available for As, Pb, Cd, chromium [(Cr(III) and Cr(VI)], mercury (Hg), nickel (Ni), Cu, Zn, and several organic compounds [25]. Thresholds for Pb in soil in children’s environments are the following:100 mg/kg: normal quality criteria;100–300 mg/kg: no action indicated;300–700 mg/kg: immediate measures should be considered and the site prioritized in the action phase;700–2500 mg/kg: immediate action should be taken.

Due to the historical use of the camp area as a shooting range, we decided to analyze the soil samples for the content of additional elements relevant to ammunition: Sb, Bi, and U. To our knowledge, limits have not been established for Sb, Bi, or U in soil.

## 3. Results

Soil concentrations for Pb, Sb, Bi, Cu, and U at the eight sampling sites and two reference sites in the Mavrovouni refugee camp are provided in Table 1.

**Lead**. While the current maximum acceptable value of Pb in soil is 100 mg/kg in children’s environments in Norway [25], five of the eight sites exceeded this threshold. Sites #3 and #7 in Mavrovouni had extreme surface soil Pb levels of approximately 8000 mg/kg and 6000 mg/kg, (relative standard deviation of 41% and 192%), respectively (Table 1). These values exceed the uppermost limit (2500 mg/kg) for soil contamination requiring immediate action [25].

**Antimony and bismuth**. The elevated concentrations of Sb (site 3: 240 mg/kg; site 7: 100 mg/kg) and Bi (site 3: 2.0 mg/kg; site 7: 3.3 mg/kg) at sites #3 and #7 strongly indicate environmental contamination (Table 1). A great variation between sampling sites in the concentrations of Pb, Sb, and Bi is illustrated in Figure 2. Maximum values for children have not been established for Sb and Bi.

**Other toxic metals**. Due to the previous military use of the area, we determined levels of elements that may occur in various types of ammunition (Table 1). Surface soil levels of As, Cd, Zn (Appendix A), and Cu did not exceed the quality thresholds specified for soil used in Norwegian kindergartens, playgrounds, and schools [25]. Compared with Sb, Bi, Cu, and Pb, the principal component analysis (PCA) clearly indicates a different source for Zn, As, and U (Figure 3). The PCA conducted on log-transformed data reduced the effect of extreme values, provided a better distinction between the groups, and explained more variation (80% versus 75%) compared with non-log-transformed data.

## 4. Comparison to Previous Report

During 2020, EAGME conducted a study of 12 soil samples and 3 water samples in Mavrovouni, published in the report “Geochemical quality control of soil and groundwater in Lesvos district” [21]. Comparing this study with the EAGME report, it is important to note the difference in (i) sampling depth and (ii) in chemicals and temperature used for digestion of soil. Sampling in the EAGME study was taken in trenches dug 10 to 15 cm deep with wood, rock fragments, leaves, etc., removed from the top layer of soil, whereas in the current study material between the sandy ground surface to about 2 cm depth was collected; the soil had no vegetation on top. Our sampling depth was similar to that of Mielke et al.’s recent study of soil Pb levels in New Orleans, sampled at 2–3 cm depth [16]. The results of the present study and the EAGME report are compared in Figure 4. In the present study (solid blue bars in Figure 4), dried (120 °C) and sieved (<2 mm) soil was microwave digested at 260 °C using a mixture of nitric (HNO_3_) and hydrofluoric (HF) acids. In the study reported by EAGME (dotted bars in Figure 4), dried (<40 °C) and sieved soil (<2 mm) was ground to <200 mesh (<0.074 mm) prior to acid digestion using a mixture of aqua regia (mixture of nitric acid and hydrochloric acid) and HF; the EAGME report gives no information on temperature.

## 5. Discussion

At the time of sampling, the investigated area was the playground for more than 2500 children [1], of whom 649 were four years old or younger; in addition, at least 118 pregnant women resided in the camp. High levels of toxic metal pollution were found in soil where children live and play in Mavrovouni. This is the first independent environmental monitoring study of the Mavrovouni refugee camp site. Although the current maximum acceptable value of Pb in soil is 100 mg/kg for playgrounds in Norway, five of eight samples from inside the camp exceeded this maximum value, with two sites containing extreme Pb concentrations (approximately 6000 and 8000 mg/kg). Sb and Bi were additionally found in surface soil and strongly indicated environmental contamination from ammunition. Previous military activity and ammunition residue on the soil surface were identified as the principal sources of the contamination. For asylum seekers in Mavrovouni, many of whom have survived violence and repeated trauma, exposure to toxic metals is a health threat on top of multiple existing crises [27,28,29,30].

Infants and children explore their environments by tasting and crawling, and carry out hand-to-mouth behavior on the ground [11]. Like all children, those living in Mavrovouni seek adventure, and a closed firing range is no exception. Infants and children typically consume a significant amount of soil and dust per day; children with pica behavior are estimated to ingest even more [12]. Levels of Pb in soil strongly influence children’s blood Pb levels [16]. Moreover, children living in residential camps can be expected to touch broken glass, fecal matter, plastic, and metal waste, in addition to soil [31].

### 5.1. Metal Toxicity

Fetuses and young children are particularly vulnerable to the toxic effects of Pb, especially their developing brains and nervous systems [32,33,34].

Pb has been used in ammunition for centuries, and Pb contamination at shooting ranges is considered a public health risk [2,35]. The soil Pb levels identified in this study fall within the ranges found by previous studies of military shooting ranges in locations ranging from the United States to South Africa [6]. Pb must be hardened before it can be used in projectiles, and the heavy metal Sb is used as an alloy.

Like Pb, Sb is toxic [36], and the combination of the two metals represents a significant pollution hazard. In 2022, the International Agency for Research on Cancer (IARC) classified Sb(III) as probably carcinogenic to humans (Group 2A) [37]. Geologically, Sb often occurs together with As and has many coincident properties. However, the principal component analysis clearly indicates a different source for As compared with Sb in the surface soil (Figure 3), and since Sb is used to increase the hardness and mechanical strength of Pb in projectiles, ammunition is the most likely source of Sb in the surface soil. Sb binds strongly to certain iron oxides and iron hydroxides [38]. However, in general, Sb has high mobility and solubility, and the mobility increases drastically with increasing pH unlike other heavy metals that dissolve at a low pH. Elevated Sb values have been observed in wildlife living in the vicinity of an Sb smelter [39]. Toxicological and exposure information is available for adults [40].

Bi shares many chemical characteristics with Pb and has been used as a replacement for Pb in shot and bullets. Compared with the high toxicity of Pb, Bi is considered a less toxic substitute to prevent accumulation of Pb in the environment [41]. Symptoms of Bi poisoning include fever, weakness, pain similar to rheumatism, and diarrhea [42].

To our knowledge, environmental limits have not been established for children’s potential exposure to Sb or Bi, which may cause potential health risks to be overlooked.

### 5.2. Choice of Reference Values

Reference values for this study are those used in Norway to assess soil in kindergartens, playgrounds, and schools for As, Cd, Cr, Cu, Pb, Hg, Ni, Zn, and several organic compounds [25] and have been used in previous research on children’s risks from polluted playground soil [43]. Principal component analysis in this study demonstrates that the most likely source for contamination of surface soil in the refugee camp at Mavrovouni is previous military activity. The reference values used here [25] do not include elements of potential military origin such as Sb and Bi. Nonetheless, the strikingly elevated concentrations of Sb and Bi observed at sites #3 and #7 are cause for concern about potential health risks for children.

### 5.3. Comparison with Previous Findings

Our study differs from the recent EAGME report [21] in terms of methods, reference values, degree of detail reported, and interpretation of the results. In the EAGME report, reference values were taken from Geochemical Atlas of Europe of the Forum of European Geological Surveys (FOREGS, now EuroGeoSurveys) that were not child-specific, and soil screening values were compared with a previous report [44]. The EAGME report noted that Greece has not established measures for measuring toxicant levels for varying types of soil use. EAGME reports that there is insufficient data to determine the geochemical background values for Lesvos. The report acknowledges that part of the area surveyed was previously used as a shooting range and is currently a reception and identification center for asylum seekers. One area had been the site of the impact of the bullets and falling shots (backstop berm) at the foot of the western-northwestern part of the hillside. EAGME considered that the sample from site MAV-12 with 330 mg/kg Pb to be “clearly” below international intervention limits; however, action limits by the Norwegian Environment Agency indicate that immediate measures should be considered at 300 mg/kg [25]. The most polluted site, where pillboxes had been used, had a sample, MAV-01, with value of 2233 mg/kg Pb but was considered “outside the site” by EAGME. EAGME writes in summary that, “the data we have do not support the existence of diffuse pollution from lead in the area of the hospitality structure in Mavrovouni, except the existence of a specific part of it with increased concentrations of lead”.

## 6. Limitations

This study does not include measurement of blood Pb levels of Mavrovouni residents or observation of the time spent in the most polluted sites. In addition, large earthworks and construction took place at the Mavrovouni refugee camp around November 2020, which may have affected soil Pb levels. Nevertheless, given the crisis situation, the vulnerable status of the residents, and the known toxicity of the contaminants, the results presented here are sufficient to mobilize immediate public health action. Ideally these children should be followed up in their eventual areas of resettlement to monitor for long-term effects and be provided comprehensive biomonitoring healthcare services, as is offered in some resettlement communities [45].

The European Commission stated in 2021 that the Mavrovouni camp “may have lacked features deemed necessary for a reception centre up to the EU standards” [46]. This study is a preliminary report that warrants further investigation to understand the extent of contamination in European refugee camps, children’s actual exposure to hazardous pollutants, and the impact on child health.

## 7. Conclusions

Children’s health is at risk, particularly in refugee camps—among other marginalized communities—where multiple risk factors intersect to create environmental injustice for the youngest [47]. Shooting ranges are known to contain dangerous levels of toxic metals, and this study shows that children in Mavrovouni are living and playing in an environment with extreme levels of Pb and other ammunition-related pollution. The most effective way to protect children from Pb is to prevent exposure in the first place. Based on the precautionary principle, the evidence presented in this study indicates the need to evacuate all children and pregnant women from the Mavrovouni refugee camp as quickly as possible.

## Figures and Tables

**Figure 1 ijerph-19-10086-f001:**
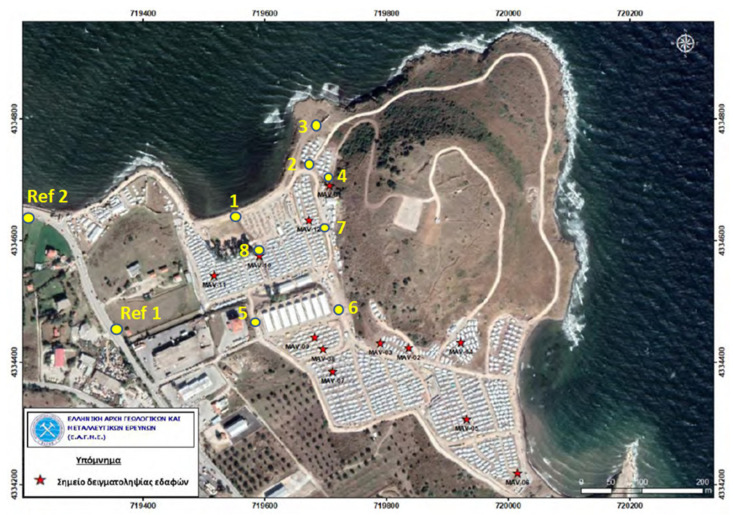
Aerial photo of the Mavrovouni camp. Yellow dots show the location of the eight sampling sites within Mavrovouni and the location for the two reference samples included in the present work. Red stars show sampling sites in the EAGME study. The small white rectangles are tents and related structures in the refugee camp where families reside. Image from EAGME [21].

**Figure 2 ijerph-19-10086-f002:**
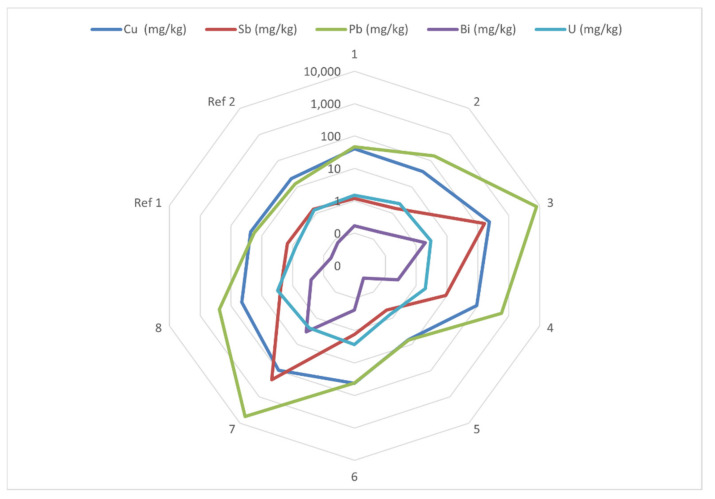
Radar chart showing the concentration (logarithmic scale; N = 7) of copper (Cu), antimony (Sb), lead (Pb), bismuth (Bi), and uranium (U) in surface soil (0–2 cm) and the variation between sampling sites. Dried (120 °C) and sieved (<2 mm) soil was microwave digested at 260 °C using a mixture of nitric (HNO_3_) and hydrofluoric (HF) acids. See Figure 1 for location of sampling sites 1–8 and reference sites Ref 1 and Ref 2.

**Figure 3 ijerph-19-10086-f003:**
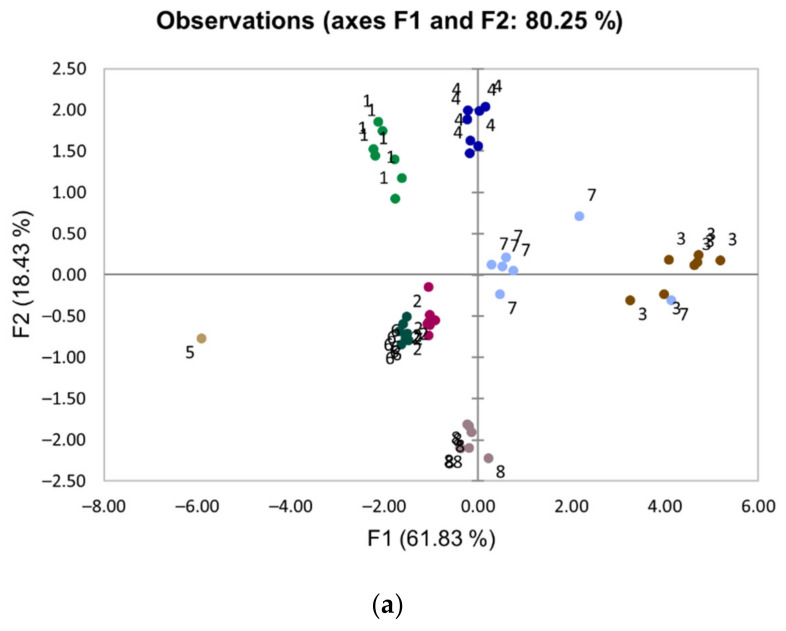
The result of the PCA analysis presented as a score plot (**a**) and a loading plot (**b**). Each sampling point in the score plot is labeled with its site number (see Figure 1). To reduce the influence of extreme values, data were log-transformed before performing PCA. The first component explains 61.83% of the variability, whereas the second component explains 18.43%. Groupings of data on the score plot may indicate two or more separate distributions in the data. Compared with the other samples, sample 5 (*n* = 1) contains large amounts of carbonates. The lengths of the arrows in the loading plot represent how well the variables explain the variation of the data. Arrows pointing in the same direction correlate positively, whereas arrows in the opposite direction correlate negatively. The load near −1 or +1 indicates that the variable strongly affected the component, and the load around zero indicates that the variable has little to no effect on the component. The loading plot clearly indicates a different source for Pb, Cu, Sb, and Bi compared with Zn, U, and As in the surface soil.

**Figure 4 ijerph-19-10086-f004:**
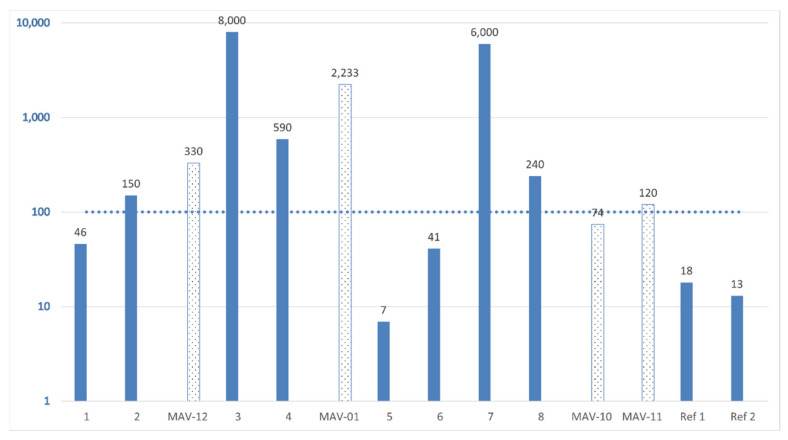
Log scale concentration (mg/kg) of lead (Pb) in surface soil sampled at 0–2 cm depth (solid blue bars) compared with Pb in soil sampled at 0.5 to 10 or 15 cm depth reported by EAGME [21]; (“MAV” values with dotted bars). See Figure 1 for location of sampling sites. The dotted line at 100 mg/kg indicates the quality criteria set for soil used in kindergartens, playgrounds, and schools in Norway [25]. Immediate action should be taken when the concentration of Pb in soil exceeds 700 mg/kg; however, immediate measures should be considered at 300 mg/kg. Abbreviations: MAV: Mavrovouni.

## Data Availability

Soil samples used in this study can be made available to other researchers by reasonable request for replication analysis. Inquiries about and requests for access to data generated and analyzed during this study should be directed to the corresponding author (K.S.).

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
