# Peer review of "Ammunition Waste Pollution and Preliminary Assessment of Risks to Child Health from Toxic Metals at the Greek Refugee Camp Mavrovouni"

_ijerph, 2022, doi:10.3390/ijerph191610086_

Round 1

Reviewer 1 Report

Dear Colleague

Thank you for giving me the opportunity to review this article

I read the article; the authors studied Ammunition waste pollution and preliminary assessment of risks to child health from toxic metals at Greek refugee camp Mavrovouni. This study is an interesting and noble work that addresses an important issue of toxic environmental. But while reading this article, there are points that the authors need to pay attention to.

1-In the introduction, the authors mention many side effect of metals. It is better use of this paper.

3-It's better to state the purpose of the study in the last paragraph of the introduction.

4-Please mention the statistical method used along with the statistical software as well as the statistical level????

5- Please fully if the word is used for the first time with aberration

6-The authors show information about method validation such as LOD, LOQ and linearity and recovery.

7- The authors show done assess the risk for toxic environmental exposures????

7-Reference format should be done according to the journal?

8-In the discussion section, Comparisons are very low. It is better for authors to compare their results with the studies of others

Best regards

Author Response

Response to Reviewer 1

Dear Colleague

Thank you for giving me the opportunity to review this article

I read the article; the authors studied Ammunition waste pollution and preliminary assessment of risks to child health from toxic metals at Greek refugee camp Mavrovouni. This study is an interesting and noble work that addresses an important issue of toxic environmental. But while reading this article, there are points that the authors need to pay attention to.

  1. In the introduction, the authors mention many side effect of metals. It is better use of this paper.

Response: We have removed 1 sentence and combined another sentence to make the section more concise.

  1. It's better to state the purpose of the study in the last paragraph of the introduction.

Response: The purpose of the study is to perform the first independent environmental monitoring study of the Mavrovouni refugee camp site on Lesvos.

  1. Please mention the statistical method used along with the statistical software as well as the statistical level

Reply: In section 2.3, we have now revised the text to state that “Principal component analysis (PCA) conducted in XLSTAT (24) was used to look for trends and commonalities in the data that can be related to properties of the different soil samples.”

  1. Please fully if the word is used for the first time with aberration

Response: We do not know which abbreviation the reviewer is referring to here. In general, we have provided all abbreviations in parentheses following the full name. The one exception to this is the reference soil NCS ZC 73007 Soil, produced by NCS Testing Technology Co., Ltd., Beijing, China; after extensive searching, we cannot find a full name for this in English. However we believe the abbreviated name is sufficient for readers to identify the specific material used.

  1. The authors show information about method validation such as LOD, LOQ and linearity and recovery.

Response: The quality of the chemical analysis is commented on in the last paragraph in chapter 2.2 with reference to Table S1. We have now added this text: “The detection and quantification limits were low compared to concentrations measured in the soil samples (Table S2, Table 2 and Table S3).” See additional information on LOD and LOQ in new Table S2.

  1. The authors show done assess the risk for toxic environmental exposures????

Response: This manuscript is a brief report offering a preliminary assessment of risk, with an aim to publish this information quickly so it is available to public health authorities. A full risk assessment and follow-up assessment are encouraged for future research.

  1. Reference format should be done according to the journal?

Response: Citations have been updated.

  1. In the discussion section, Comparisons are very low. It is better for authors to compare their results with the studies of others

Response: We have included comparisons to multiple other settings, such as toxic metal contaminant levels in Norwegian playgrounds (color coding in Table 2), where the Mavrovouni samples were extremely high. In addition we found that soil lead levels identified in this study fall within the ranges found by previous studies of military shooting ranges in locations ranging from the United States to South Africa (6), showing that this was a ’typical’ shooting range in terms of lead contamination. To our knowledge, relevant comparisons for bismuth or antimony were not available and have stated this in the discussion.

Reviewer 2 Report

The manuscript titled “Ammunition waste pollution and preliminary assessment of risks to child health from toxic metals at Greek refugee camp Mavrovouni” presents an interesting assessment of the exposure to heavy metals and metalloids of subjects residing in a refugee camp. The manuscript is in general well written and clear. Some minor revisions are suggested to improve the quality of the reporting.

Please check and revise the formatting of bibliography to be revised according to journal style, with references numbered in the order they are cited in the text.

Please note that a maximum of ten keywords is allowed.

In the Results, authors present that surface soil levels of some elements did not exceed the quality threshold (L201-202) and data are not shown, but below in the text zinc and arsenic are assessed in the PCA analysis. As consequence, it is suggested to present all data, at least in supplementary material, avoiding the statement ‘data not shown’.

In Table 2, the first row seem missing of one element, please check and revise.

Not mandatory, but it may be interesting if subjects are allowed to grow and consume vegetables or other food inside the camp, thus an additional source of exposure may be intake of contaminated foods. Authors discussed that the study area is used as playground, but considering the map presented, it may be interesting to explore this possible additional source of exposure.

Author Response

Response to Reviewer 2

The manuscript titled “Ammunition waste pollution and preliminary assessment of risks to child health from toxic metals at Greek refugee camp Mavrovouni” presents an interesting assessment of the exposure to heavy metals and metalloids of subjects residing in a refugee camp. The manuscript is in general well written and clear. Some minor revisions are suggested to improve the quality of the reporting.

  1. Please check and revise the formatting of bibliography to be revised according to journal style, with references numbered in the order they are cited in the text.

Response: Thank you; we have updated the citations accordingly.

  1. Please note that a maximum of ten keywords is allowed.

Response: We have now removed several keywords and hope that the editors are willing to allow an extra 3 keywords in Greek that will allow this article to be more findable for Greek readers.

  1. In the Results, authors present that surface soil levels of some elements did not exceed the quality threshold (L201-202) and data are not shown, but below in the text zinc and arsenic are assessed in the PCA analysis. As consequence, it is suggested to present all data, at least in supplementary material, avoiding the statement ‘data not shown’

Response: Thank you for this comment. We have now added additional information on arsenic, cadmium, and zinc in new Table S3.

  1. In Table 2, the first row seem missing of one element, please check and revise.

Response: We apologize for the cut-and-paste error; we have reformatted the table so that uranium is now visible as the last column heading.

  1. Not mandatory, but it may be interesting if subjects are allowed to grow and consume vegetables or other food inside the camp, thus an additional source of exposure may be intake of contaminated foods. Authors discussed that the study area is used as playground, but considering the map presented, it may be interesting to explore this possible additional source of exposure.

Response: We agree that there are many opportunities to follow up the refugee camp residents more closely to better understand the impact and possible mitigating factors of exposure to toxic metals. As this study focused on soil only and did not have approval to study human subjects, we encourage additional research teams to pursue the questions raised by the reviewer.  

Reviewer 3 Report

I have put a few comments in text that should be considered.

Author Response

Response to Reviewer 3

I have put a few comments in text that should be considered.

  1. Abstract – “both studies are post-refugee camp. replace "prior to" with "subsequent to"”

Response: Thank you; this has been revised.

  1. 4 – “should be <100 ppm”

Response: 100 mg/kg is the suggested threshold for normal concentrations of lead in surface soil.

  1. 4 – “but elevated concentrations of these elements may help in differentiating anthropogenic lead associated with the shooting range relative to natural background samples.”

Response: We agree with the reviewer. In the manuscript, we argue that the concentration of antimony and bismuth in surface soil strongly indicate environmental contamination, most likely from previous military activity and ammunition residue.  

  1. Table 2 – “samples 1, 5, 6 and the two ref samples should be green...”

Response: Thank you; these have been made green, along with the copper findings which were below their respective action limits.

  1. Discussion – last paragraph “all European refugee camps, or this one built on a former shooting range? This study does not support a wholesale review of all refugee camps.”

Response: We have now moved text from the introduction to this section, to call attention to the European Commission’s statement that Mavrovouni “may have lacked features deemed necessary for a reception centre up to the EU standards.” While the study does not call for a “wholesale review of all refugee camps,” it does shine a spotlight on the gaps in the standards in the EC’s refugee system that facilitated the relocation of refugees to this site just hours after it had been used for military shooting training. This is more relevant now than ever as over 5 million Ukrainian refugees have entered Europe in 2022; UNICEF estimates that approximately half of these are children.

  1. Conclusion – “it also argues for a more comprehensive study/risk assessment to understand the exposure and possible risk mitigation measures to reduce such exposure (e.g., capping, fencing or certain areas). immediate evacuation will cause other issues...”

Response: Asylum-seekers in Mavrovouni continue to face trauma, high risk of violence including suicide, lack of services, stress, legal and political limbo, and hopelessness. Based on extensive work on site with refugees in Mavrovouni, we the authors do call for evacuation of refugees from this camp. UNICEF and other humanitarian actors have called for all children to be transferred off Lesvos. In addition, we encourage long-term follow-up studies of the affected children, including in their eventual places of resettlement.

This manuscript is a resubmission of an earlier submission. The following is a list of the peer review reports and author responses from that submission.

Round 1

Reviewer 1 Report

The report highlights the existence of lead contaminated soil at a refugee camp and potential exposure to children based on a limited number of soil samples. The conclusions should state that further investigation is warranted to understand the degree and extent of contamination, and the potential exposure and impact to children - this is a cursory report which definitely indicates the need for such work.  

Why so few samples?  Why not use a portable XRF for screening? Report spends was too much time on testing methodology.

Report indicates samples collected where children spend their time, however with no support for this. Large areas of camp with no samples...  This is an important element from an exposure standpoint.

It cites another study of soil samples from same camp, but spends more time  discounting data than using it. Figure 4 only shows selected results from EAGME - why?  Any samples above their reference (background) samples should be cited as indication of contamination and need for further investigation.  Notably, lead in all all 12 EAGME samples were above your R1 and R2...

Was EAGME not independent?

Figure 3 is confusing. The presence lead and related metals above the reference values of R1 and R2 at a former shooting range is sufficient in my opinion.

Report makes no mention of ammunition-related detritus found at ground surface any residents.

Report fails to indicate extensive earthworks done at site for construction of camp and later to control flooding - this would affect distribution of lead contamination and should be cited as factor especially in tented areas (where many of EAGME samples were taken and undoubtedly why most were relatively low).

Conclusions should be stronger, highlighting the need for more work or relocation!

Author Response

Response to Reviewer 1

Reviewer 2 Report

The manuscript is written outstanding and describes the current issue, risk to child health from toxic metals at Greek refugee camp which is very interesting. 

I have just a few minor comments. 

I wonder how long this place had been used for military shooting range before "unused". Depending on how many bullets were fired / depending on the time being used as the range, the contamination level would change. It would help the readers if the authors can show the comparison with other places which has been used as military range.  

There must be some other data for soil contamination of other military shooting range in the world. It would be helpful if the authors can add other countries' as well. 

Author Response

Response to Reviewer 2

Reviewer 3 Report

The authors have reported on heavy metals in the soil of a refugee camp. This is very important information for the relevant agency/ies that regulate these camps but I do not see what the relevance is for an international health journal. The research does not advance our understanding of the science, either environmental contamination or the health effects of these contaminants. There is no investigation of actual exposure or health - the authors have acknowledged this is a limitation

I have minimal concerns with the methodology, although there is no indication of whether the soil sampling was conducted using an accepted/validated method. This was not mentioned in either the methods section or when the authors compared their results to a previous investigation (EAGME).

Author Response

Response to Reviewer 3
